A machine learning approach for identification of gastrointestinal predictors for the risk of COVID-19 related hospitalization

http://orcid.org/0000-0001-8257-8567 Lipták Peter 1 peter.liptak@uniba.sk
Banovcin Peter 1
http://orcid.org/0000-0003-0429-6833 Rosoľanka Róbert 2
Prokopič Michal 1
Kocan Ivan 3
Žiačiková Ivana 3
Uhrik Peter 1
http://orcid.org/0000-0002-6712-3457 Grendar Marian 4 5
Hyrdel Rudolf 1
1 Gastroenterology Clinic, University Hospital in Martin, Jessenius Faculty of Medicine in Martin, Comenius University in Bratislava , Martin , Slovak Republic
2 Clinic of Infectology and Travel Medicine, University Hospital in Martin, Jessenius Faculty of Medicine in Martin, Comenius University in Bratislava , Martin , Slovak Republic
3 Clinic of Pneumology and Phthisiology, University Hospital in Martin, Jessenius Faculty of Medicine in Martin, Comenius University in Bratislava , Martin , Slovak Republic
4 Laboratory of Bioinformatics and Biostatistics, Biomedical Centre Martin, Jessenius Faculty of Medicine, Comenius University in Bratislava , Martin , Slovak Republic
5 Laboratory of Theoretical Methods, Institute of Measurement Science, Slovak Academy of Sciences , Bratislava , Slovakia
Suner Aslı
Electronic publication date: 2022 Mar 21
Publication date: 2022
Volume: 10
Electronic Location ID: e13124
Received 2021 Sep 1; Accepted 2022 Feb 24
Copyright: © 2022 Lipták et al.
Copyright year: 2022
Copyright holder: Lipták et al.
License: This is an open access article distributed under the terms of the Creative Commons Attribution License, which permits unrestricted use, distribution, reproduction and adaptation in any medium and for any purpose provided that it is properly attributed. For attribution, the original author(s), title, publication source (PeerJ) and either DOI or URL of the article must be cited.
License URL: https://creativecommons.org/licenses/by/4.0/

Keywords: COVID-19, SARS-CoV-2, Machine learning, Artificial intelligence, Random forest, Symptoms, Liver, Predictors, Hospitalization

Funding: Research and Development of Telemedicine Solutions to Support the Fight against Pandemic Diseases Induced COVID-19 and Reducing its Negative Consequences by Monitoring the Health Status of People in Order to Eliminate the Risk of Infection in at-risk Populations, ITMS 313011ASY8 New Possibilities for Laboratory Diagnostics and Massive Screening of SARS-Cov-2 and Identification of Mechanisms of Virus Behavior in Human Body, ITMS 313011AUA4 Ministry of Health of the Slovak Republic 2019/44-UKMT-7 This publication has been produced with the support of: The Integrated Infrastructure Operational Program for the project: Research and development of telemedicine solutions to support the fight against pandemic diseases induced COVID-19 and reducing its negative consequences by monitoring the health status of people in order to eliminate the risk of infection in at-risk populations, ITMS: 313011ASY8, co-financed by the European Regional Development Fund, by the Integrated Infrastructure Operational Program for the project: New possibilities for laboratory diagnostics and massive screening of SARS-Cov-2 and identification of mechanisms of virus behavior in human body, ITMS: 313011AUA4, co-financed by the European Regional Development Fund; and by Ministry of Health of the Slovak Republic under the project registration number 2019/44-UKMT-7. The funders had no role in study design, data collection and analysis, decision to publish, or preparation of the manuscript.

==============================
Background and aim

COVID-19 can be presented with various gastrointestinal symptoms. Shortly after the pandemic outbreak, several machine learning algorithms were implemented to assess new diagnostic and therapeutic methods for this disease. The aim of this study is to assess gastrointestinal and liver-related predictive factors for SARS-CoV-2 associated risk of hospitalization.

Methods

Data collection was based on a questionnaire from the COVID-19 outpatient test center and from the emergency department at the University Hospital in combination with the data from internal hospital information system and from a mobile application used for telemedicine follow-up of patients. For statistical analysis SARS-CoV-2 negative patients were considered as controls in three different SARS-CoV-2 positive patient groups (divided based on severity of the disease). The data were visualized and analyzed in R version 4.0.5. The Chi-squared or Fisher test was applied to test the null hypothesis of independence between the factors followed, where appropriate, by the multiple comparisons with the Benjamini Hochberg adjustment. The null hypothesis of the equality of the population medians of a continuous variable was tested by the Kruskal Wallis test, followed by the Dunn multiple comparisons test. In order to assess predictive power of the gastrointestinal parameters and other measured variables for predicting an outcome of the patient group the Random Forest machine learning algorithm was trained on the data. The predictive ability was quantified by the ROC curve, constructed from the Out-of-Bag data. Matthews correlation coefficient was used as a one-number summary of the quality of binary classification. The importance of the predictors was measured using the Variable Importance. A 2D representation of the data was obtained by means of Principal Component Analysis for mixed type of data. Findings with the p-value below 0.05 were considered statistically significant.

Results

A total of 710 patients were enrolled in the study. The presence of diarrhea and nausea was significantly higher in the emergency department group than in the COVID-19 outpatient test center. Among liver enzymes only aspartate transaminase (AST) has been significantly elevated in the hospitalized group compared to patients discharged home. Based on the Random Forest algorithm, AST has been identified as the most important predictor followed by age or diabetes mellitus. Diarrhea and bloating have also predictive importance, although much lower than AST.

Conclusion

SARS-CoV-2 positivity is connected with isolated AST elevation and the level is linked with the severity of the disease. Furthermore, using the machine learning Random Forest algorithm, we have identified the elevated AST as the most important predictor for COVID-19 related hospitalizations.

Introduction

Acute SARS-CoV-2 infection presents with variable symptoms associated with various organ systems. Typical symptoms of COVID-19 are fever, cough, and in the case of a more severe course of the disease, dyspnea with respiratory insufficiency occurs (Guan et al., 2020). In addition, COVID-19 may be presented with gastrointestinal symptoms, which include dominantly nausea, vomiting, diarrhea, anorexia and abdominal pain with relatively wide range of prevalence among different published studies (Aziz et al., 2020; Mao et al., 2020; Sultan et al., 2020; Patel et al., 2020; D’Amico et al., 2020; Jin et al., 2020; Xiao et al., 2020). Since COVID-19 pandemic is the cause of an immense world health crisis, new diagnostic and therapeutic methods are rapidly emerging (Alimadadi et al., 2020). The use of artificial intelligence is just one of them. Shortly after the COVID-19 outbreak, various machine learning algorithms have been implemented (Randhawa et al., 2020; Yan et al., 2020; Ge et al., 2021; Li et al., 2020). Machine learning helps quickly identify patterns and trends of the large volume of data, that are difficult for humans to recognize (Kushwaha et al., 2020). The availability of objective stratification tools for the rapid assessment of a patient status and prognosis is of great use for the frontline health-care providers (Bachtiger, Peters & Walsh, 2020).

The primary aim of this study is to assess the possible predictive factors for SARS-CoV-2 outcome based on gastrointestinal symptoms and liver related laboratory results using machine learning algorithms of the Random Forest (Guan et al., 2020; Breiman, 2001). The secondary aim is to determinate the prevalence of gastrointestinal symptoms among patients with COVID-19 within different groups based on the severity of the disease.

Materials and Methods

The study was performed from February through May 2021. Only subjects aged 18 years or older were included in the study. All patients enrolled in this study had signed the informed consent.

This study was approved by the Ethical Committee of the University Hospital in Martin, decision number: 14/2021.

Two distinct kinds of population were considered for this study. First group consists of patients who underwent nasopharyngeal swab in the outpatient hospital testing center for COVID-19 in order to determine whether they were SARS-CoV-2 positive. The method used for SARS-CoV-2 detection from nasopharyngeal swab was PCR (polymerase chain reaction). This group was then subdivided based on their positivity. The negative group was thereafter used as a control group for this study.

Second group consists of patients who attended COVID-19 emergency department (ED) in the University Hospital. These patients were confirmed positive from nasopharyngeal swab either by PCR or antigen method. Only patients with typical COVID-19 symptoms (fever, cough, dyspnoe) were included in this study. Patients who were SARS-CoV-2 positive but, didn’t present with typical COVID-19 symptoms (e.g., patients who came to emergency room because of other diagnoses, but simultaneously were SARS-CoV-2 positive) were excluded. Therefore, we considered for this study only patients who were both tested positive and had at least one typical COVID-19 symptom.

The second group was then divided based on further evaluation and course of the disease. First subgroup consists of patients that didn’t require admission to the hospital and were referred to the outpatient care. Second subgroup of patients was admitted to the hospital. Consequently, this group was observed until the end of hospitalization either because of their death or resolution of the disease. This subgroup was also divided for analysis purposes to patients who required medical care in standard hospital ward and those who needed intensive care unit (ICU).

Data was collected by using a questionnaire in the group from COVID-19 outpatient test center at the University Hospital. Data from emergency room was obtained with the same questionnaire which was combined with information from medical examination by an attending physician and from the mobile application MEDAsistent used for telemedicine follow-up developed at the Clinic of Pneumology and Phthisiology in the University Hospital in Martin. Further information (including laboratory tests results, chest X-ray etc.) about patients who were hospitalized has been obtained from hospital information system.

The questionnaire consists of questions related to the present health complaints typical for COVID-19 and the spectrum of most common gastrointestinal symptoms which occurred in the last 5–7 days before examination. Patients were also allowed to write down other presented symptoms in the case they were not in the original list. In order to include only new or worsened gastrointestinal symptoms in the study the questionnaire also contained questions about chronic gastrointestinal symptoms and their possible worsening in the last 5–7 days before examination.

Data analysis

The data was visualized and analyzed in R (R Development Core Team, 2021), version 4.0.5, with the aid of the libraries gtsummary (Sjoberg et al., 2020), rstatix (Kassambara, 2021), DescTools (Signorell, 2021), randomForestSRC (Ishwaran & Kogalur, 2021), PCAmixdata (Chavent et al., 2017) and ggpubr (Kassambara, 2020). The sample median and the lower and upper quartiles were used to summarize the data on continuous variables (e.g., age); counts and percentages were used to summarize factors (e.g., gender). The Chi-squared or Fisher test were applied to test the null hypothesis of independence between factors (gender vs group; fever vs group; cough vs group; diarrhea vs group; constipation vs group; bloating vs group; nausea vs group; heartburn vs group; abdominal pain vs group), followed, where appropriate, by multiple comparisons with the Benjamini Hochberg adjustment. Using a contingency table, an absence of trend was tested by Cochran Armitage test. The null hypothesis of the equality of the population medians of the continuous variable: age, Oxygen (O2) saturation, C-reactive protein (CRP), gamma glutamyltransferase (GMT), aspartate aminotrasferase (AST), Bilirubin) was tested by the Kruskal Wallis test, followed by the Dunn multiple comparisons test with the Benjamini Hochberg correction of p-values. Two-way ANOVA was used to model the association between AST and group (discharged home, admitted to hospital) in interaction with the recent ATB use. Another two-way ANOVA model was utilized to quantify the association between AST and group (discharged home, admitted to hospital) in interaction with chronic liver disease (yes, no). The AST values were log-transformed to bring data to normality. Normality of residuals was assessed by the quantile-quantile plot with the 95% confidence band constructed by bootstrap. Assumption of homogenity of variance was tested by the Levene test. In order to assess the predictive power of the gastrointestinal parameters and other measured variables (Gender, Age, No of Days of Symptoms, AST, alanine aminotrasferase /ALT/, Bilirubin, Recent antibiotics /ATB/ Usage, Diabetes Mellitus, Arterial Hypertension, Chronic Liver Disease, Fever, Cough, Diarrhea, Constipation, Bloating, Nausea, Heartburn, Abdominal Pain) for predicting the outcome of the patient group the Random Forest machine learning algorithm was trained on the data. The predictive ability was quantified by the ROC curve, constructed from the Out-of-Bag data. The Matthews correlation coefficient was used as a one-number summary of the quality of binary classification. Importance of the predictors was measured by the Variable Importance. A 2D representation of the data (the predictors used in Random Forest; i.e., Gender, Age, Number of Days of Symptoms, AST, ALT, Bilirubin, Recent ATB Usage, Diabetes Mellitus, Arterial Hypertension, Chronic Liver Disease, Fever, Cough, Diarrhea, Constipation, Bloating, Nausea, Heartburn, Abdominal Pain) was obtained by Principal Component Analysis for a mixed type of data. Findings with the p-value below 0.05 were considered statistically significant.

Results

A total of 710 patients were enrolled in the study. Thirty (30) patients were excluded from the further analysis after primary screening. Participants (n = 352) from the outpatient center who were tested PCR negative for SARS-CoV-2 virus were considered as the control group. SARS-CoV-2 positive group from outpatient center included 166 participants. One hundred and sixty-two (n = 162) patients from emergency department were enrolled. From this group 78 patient (48%) were discharged home, 57 (35.3%) admitted to the hospital for standard care until discharged from hospital. Twenty-seven (27) (16.7%) patients required intensive care unit. Based on age, the groups from outpatient center had almost similar median of 42 and 41 years of age respectively. Hospitalized patients were significantly older as shown in the Table 1. The presence of typical COVID-19 symptoms such as fever and cough were significantly higher in the hospitalized groups as opposed to outpatient participants. There were no significant differences between groups in the men to women ratio.

Table 1 Patients characteristics.

		Outpatient test center (SARS-CoV-2 negative) (n = 352)1	Outpatient test center (SARS-CoV-2 positive) (n = 166)1	Discharged home (n = 78)1	Admitted to hospital-standard care (n = 57)1	Admitted to hospital-ICU (n = 27)1	p value2	
Age		42 (31, 51)	41 (32, 52)	48 (42, 58)	55 (46, 67)	65 (55, 72)	<0.001	
Gender							0.07	
	female	206 (59%)	104 (63%)	41 (53%)	27 (47%)	11 (41%)		
	male	141 (40%)	62 (37%)	37 (47%)	30 (53%)	16 (59%)		
	NA	5 (1%)						
Fever							<0.001	
	Not-presented	340 (97%)	146 (88%)	48 (62%)	25 (44%)	15 (56%)		
	Presented	12 (3.4%)	20 (12%)	30 (38%)	32 (56%)	12 (44%)		
Cough							<0.001	
	Not-presented	342 (97%)	144 (87%)	48 (62%)	24 (42%)	14 (52%)		
	Presented	10 (2.8%)	22 (13%)	30 (38%)	33 (58%)	13 (48%)		
Diarrhea							<0.001	
	Not-presented	330 (94%)	141 (85%)	64 (82%)	34 (60%)	19 (70%)		
	Presented	22 (6.2%)	25 (15%)	14 (18%)	23 (40%)	8 (30%)		
Constipation							0.37	
	Not-presented	347 (99%)	164 (99%)	76 (97%)	55 (96%)	26 (96%)		
	Presented	5 (1.4%)	2 (1.2%)	2 (2.6%)	2 (3.5%)	1 (3.7%)		
Bloating							0.13	
	Not-presented	341 (97%)	161 (97%)	76 (97%)	51 (89%)	26 (96%)		
	Presented	11 (3.1%)	5 (3.0%)	2 (2.6%)	6 (11%)	1 (3.7%)		
Nausea							<0.001	
	Not-presented	340 (97%)	147 (89%)	70 (90%)	45 (79%)	23 (85%)		
	Presented	12 (3.4%)	19 (11%)	8 (10%)	12 (21%)	4 (15%)		
Heart burn							0.86	
	Not-presented	341 (97%)	163 (98%)	76 (97%)	56 (98%)	26 (96%)		
	Presented	11 (3.1%)	3 (1.8%)	2 (2.6%)	1 (1.8%)	1 (3.7%)		
Abdominal pain							0.07	
	Not-presented	334 (95%)	150 (90%)	71 (91%)	49 (86%)	26 (96%)		
	Presented	18 (5.1%)	16 (9.6%)	7 (9.0%)	8 (14%)	1 (3.7%)		
Notes:

1 Statistics presented: n (%); Median (IQR).

2 Statistical tests performed: Kruskal-Wallis test; Fisher’s Exact Test for Count Data with simulated p-value (based on 2,000 replicates).

Gastrointestinal symptoms occurrence and laboratory findings (Tables 1 and 2)

The presence of diarrhea, constipation, bloating, nausea, heartburn and abdominal pain was considered in this study. Presence of diarrhea and nausea was significantly higher in SARS-CoV-2 positive patients than in SARS-CoV-2 negative controls. Comparing SARS-Cov-2 negative and SARS-CoV-2 positive participants the cumulative presence of diarrhea is 21.3% (70/328) in the positive group (combined outpatient center and emergency department) vs 6.2% (22/352) in the negative group and for nausea it is 13.1% (43/328) in the positive group vs 3.4% (12/352) in the negative group. This trend goes further considering ED patients and the severity of disease.

Table 2 Gastrointestinal symptoms occurrence and laboratory findings.

		Discharged home (n = 78)1	Admitted to hospital-standard care (n = 57)1	Admitted to hospital-ICU (n = 27)1	p value2	
Gender					0.55	
	female	41 (53%)	27 (47%)	11 (41%)		
	male	37 (47%)	30 (53%)	16 (59%)		
No of days with symptoms		7 (5,10)	7 (5.10)	5 (2,7)	0.03	
Age		48 (42, 58)	55 (46, 67)	65 (55, 72)	<0.001	
Diarrhea					0.02	
	Not-presented	64 (82%)	34 (60%)	19 (70%)		
	Presented	14 (18%)	23 (40%)	8 (30%)		
Abdominal pain					0.23	
	Not-presented	71 (91%)	48 (84%)	26 (96%)		
	Presented	7 (9.0%)	9 (16%)	1 (3.7%)		
Vomitus					0.24	
	Not-presented	65 (83%)	53 (93%)	23 (85%)		
	Presented	13 (17%)	4 (7.0%)	4 (15%)		
Nausea					0.22	
	Not-presented	70 (90%)	45 (79%)	23 (85%)		
	Presented	8 (10%)	12 (21%)	4 (15%)		
Heart burn					0.83	
	Not-presented	76 (97%)	56 (98%)	26 (96%)		
	Presented	2 (2.6%)	1 (1.8%)	1 (3.7%)		
Bloating					0.03	
	Not-presented	76 (97%)	49 (86%)	26 (96%)		
	Presented	2 (2.6%)	8 (14%)	1 (3.7%)		
Constipation					>0.9	
	Not-presented	76 (97%)	55 (96%)	26 (96%)		
	Presented	2 (2.6%)	2 (3.5%)	1 (3.7%)		
O2 saturation (%)		96 (95, 98)	93 (89, 96)	87 (83, 91)	<0.001	
CRP (norm < 5 mg/l)		29 (8, 74)	99 (65, 140)	102 (56, 166)	<0.001	
AST (norm < 0.6 ukat/l)		0.56 (0.41, 0.72)	0.74 (0.60, 0.97)	0.91 (0.68, 1.41)	<0.001	
ALT (norm < 0.6 ukat/l)		0.48 (0.35, 0.88)	0.61 (0.45, 0.80)	0.61 (0.36, 1.21)	0.25	
Bilirubin (norm < 21 ukat/l)		8.8 (7.1, 11.1)	9.2 (8.3, 13.9)	9.5 (7.8, 12.5)	0.25	
Recent ATB usage					0.09	
	No	45 (58%)	22 (39%)	13 (48%)		
	Yes	33 (42%)	35 (61%)	14 (52%)		
Diabetes Mellitus					<0.001	
	No	72 (92%)	45 (79%)	15 (56%)		
	Yes	6 (7.7%)	12 (21%)	12 (44%)		
Arterial Hypertension					0.003	
	No	49 (63%)	26 (46%)	7 (26%)		
	Yes	29 (37%)	31 (54%)	20 (74%)		
Chronic liver disease	No	72 (92%)	47 (82%)	22 (81%)		
	Yes	6 (8%)	10 (18%)	5 (19%)		
Notes:

1 Statistics presented: n (%); Median (IQR).

2 Statistical tests performed: chi-square test of independence; Kruskal-Wallis test; Fisher’s exact test.

CRP, C- reactive protein; AST, aspartate transaminase; ALT, alanine transaminase; ATB: antiobiotics.

Among gastrointestinal symptoms, diarrhea and bloating were significantly more often manifested in patients who were admitted to the hospital compared to those discharged home (40% for diarrhea and 14% for bloating vs 18% and 2.6% respectively). Other symptoms such as abdominal pain, heart burn, nausea, vomitus, anorexia, and constipation were not presented differently in these groups in the meaning of statistical significance. C-reactive protein was also significantly higher in hospitalized group. In case of alanin transaminase (ALT), aspartate transaminase (AST) and bilirubin as markers of possible liver damage only AST (Fig. 1) was significantly higher in the hospitalized group. This difference is substantial. There is no statistically significant difference in the levels of ALT (Fig. 1) and Bilirubin when comparing different groups of patients.

Figure 1 Aspartate and alanine transaminase in hospitalized patients vs discharged home.

AST, Aspartate transaminase; ALT, Alanine transaminase; p < 0.001. AST activity is significantly higher in the hospitalized group compared to patients discharged home after visit to emergency department. There are no significant differences in ALT activity between this groups of patients.

Predictors of hospitalization based on machine learning

Based on the Random Forest algorithm with the data on demographic characteristics, symptoms and gastrointestinal related laboratory findings in hospitalized and discharged patients, several predictors for risk of hospitalization were identified. AST was pinpointed as the most important predictor followed by age and diabetes mellitus. Diarrhea and bloating have also positive importance, although much lower than AST. Gastrointestinal symptoms such as nausea, abdominal pain or anorexia have none or negative predictive importance. The ROC curve for combined factors is shown in the Fig. 2 with AUC 0.76. The Matthews correlation coefficient was 0.48.

Figure 2 The ROC curve for general COVID-19 and gastrointestinal symptoms and other measurable data in general clinical settings.

Out-of-bag receiver operating characteristic curve with calculated area under the curve (AUC) = 0.76. The Matthews correlation coefficient was 0.48. For analysis were considered: general COVID-19 symptoms, gastrointestinal symptoms, age, sex, lasting of the symptoms and comorbidities (diabetes mellitus, arterial hypertension and chronic liver diseases).

When using only liver enzymes (AST, ALT), gastrointestinal symptoms (diarrhea and bloating), chronic liver disease, age and diabetes mellitus, the ROC curve (Fig. 3) for this combination of factors attained AUC 0.799 with AST as the strongest predictor for hospitalization (Table 3). The Matthews correlation coefficient was 0.37.

Figure 3 The ROC curve for selected parameters.

Out-of-bag receiver operating characteristic curve with calculated area under the curve (AUC) = 0.799. The Matthews correlation coefficient was 0.37. For analysis were considered selected parameters (clinically easily measurable): liver enzymes (AST, ALT), gastrointestinal symptoms (diarrhea and bloating), chronic liver disease, age and diabetes mellitus.

Table 3 Relative predictors values.

		Variable importance	
	All	Discharged home	Admitted to hospital	
AST	0.1451	0.5729	0.2217	
Diabetes Mellitus	0.0248	0.1080	0.0288	
Chronic liver disease	0.0169	0.0882	0.0061	
ALT	0.0110	0.1025	−0.0379	
Diarrhea	0.0068	0.0087	0.0272	
Age	0.0139	0.0561	0.0197	
Bloating	0.0011	0.0052	0.0006	
Note:

AST, aspartate transaminase; ALT, alanine transaminase.

Principal components visualization of data

Principal component analysis was used to get a two-dimensional visualization of the data, for patients discharged home after ED examination and patients admitted to hospital. Data used for the analysis consist of the data from Table 2, that means a combination of general patient characteristics, typical COVID-19 symptoms and gastrointestinal symptoms and liver related laboratory results. The PCA plot (Fig. 4) is showing two distinct clusters which are partially overlapping with tendencies to shift apart.

Figure 4 Principal component analysis for mixed type of data to obtain two-dimensional representation of the data.

Patients who were discharged home are marked as black dots and those who were admitted to the hospital marked as red dots. The first principal component (x axis) explains 14.04% of the variability; the second principal component (y axis) explains 10.44% of the variability in data. The two groups cannot be completely separated, as there is some overlap of the observations but there is a clear tendency to shift apart of the clusters.

Discussion

Several studies and meta-analyses have pointed out the gastrointestinal involvement in the SARS-CoV-2 infection (Mao et al., 2020; Sultan et al., 2020; D’Amico et al., 2020; Xiao et al., 2020; Pan et al., 2020; Villapol, 2020; Galanopoulos et al., 2020). The data from the pooled prevalence of gastrointestinal symptoms are varying significantly from 10.5% to 53% between studies (Mao et al., 2020; Sultan et al., 2020; Pan et al., 2020; Ashktorab et al., 2021). Based on comprehensive meta-analysis by Sultan et al. (2020), the pooled prevalence of diarrhea is 7.7%, nausea and vomiting 7.8% and abdominal pain 3.6%. In the presented study we have focused on the presence of diarrhea, constipation, bloating, nausea, heart burn and abdominal pain. Statistically significant differences have been found in the case of diarrhea and nausea when comparing SARS-CoV-2 negative and positive patients. In the group of hospitalized patients (with standard care) the diarrhea was presented in 40% patients and nausea in 21%, which is higher compared to some meta-analysis mentioned, but consistent with the data considering general presence of gastrointestinal symptoms and gut involvement. When comparing only emergency department group the presence of bloating is significantly higher in the hospitalized group than in those who were discharged home. Interestingly, bloating has lower prevalence in the group of ICU patients than in patients with standard care management. This could be explained by high subjectivity and interpersonal differences when reporting symptom such as bloating. Considering differences between these two groups of patients, those with more severe course of disease attach lower importance to less annoying symptoms such as heart burn, bloating and nausea when compared to more manifested symptoms such as diarrhea, abdominal pain or vomitus.

Focusing on the liver enzymes as markers of possible liver impairment resulting from SARS-CoV-2 infection the AST, ALT and bilirubin were considered for the evaluation. The results are showing that median level of liver enzymes was not elevated in the discharged group. Bilirubin and ALT were also within normal range in the hospitalized group with no statistically significant differences between these two groups. Only AST was elevated over the upper level of the reference value in the hospitalized group with progressively higher values in patients who required ICU. The differences between hospitalized and discharged patients are substantially significant.

Several previously published data have shown an elevation in both transaminases and bilirubin to a different extent ranging from 1% to 53% (mainly ALT and AST accompanied by slightly increased bilirubin concentrations) (Mao et al., 2020). In most published data, severe liver alterations were uncommon (Marasco et al., 2021) and the pooled prevalence of liver injury regarding severity was 12% based on the meta-analysis by Mao et al. (2020). More severe liver injury was also associated with worse outcomes, including intensive care unit admission and mortality (Phipps et al., 2020).

The pathophysiology of liver involvement in COVID-19 is still not completely understood. The direct viral infection of the liver cells is proposed as one of potential causes of liver injury, but the comprehensive studies are scarce. A study with pathological analysis of liver tissues from dead victims of COVID-19 showed no viral inclusions in hepatocytes (Zhang, Shi & Wang, 2020). Another repeatedly proposed and generally accepted mechanism of liver impairment could be drug toxicity (Mao et al., 2020). In order to determine the possible influence of recent ATB usage on the elevation of AST presented in this paper, a two-way analysis of variance (two-way ANOVA) was performed. There are no significant differences between the groups with or without recent antibiotics usage. Therefore, we have concluded that ATB usage has no relevant influence on the elevated AST levels. The two-way ANOVA was also performed to assess the relationship between the presence of chronic liver disease and AST. There is no statistically relevant difference in AST levels in hospitalized patients with and without chronic liver disease.

Another possible explanation of elevated transaminases is that it could be the result of a systemic inflammation. ALT is an enzyme most commonly found in liver, with small levels in striated muscle tissue and myocardium. On the other hand, AST could be found in liver, but also in striated and myocardial muscle, kidneys, brain and red blood cells. AST had been used as a marker for myocardial infarction for a long time before more sensitive markers were identified and implemented into the routine clinical practice (Ndrepepa, 2021). Based on the results of this study and current knowledge of SARS-CoV-2 interaction in human organism it is possible that elevated levels of AST in COVID-19 patients could be the result of a systemic inflammation with general tissue hypoperfusion rather than a result of a direct influence of SARS-CoV-2 virus on the hepatocytes or hepatotoxic drug use.

Further, we focused on identifying the possible predicting factors for hospitalization in COVID-19 patients using the Random Forest (RF) machine learning algorithm.

Different types of machine learning are being used in an increased rate to determine the predictors of outcome in various areas of clinical practice from brain trauma injuries (Hanko et al., 2021), radiology (Choy et al., 2018), oncology (Cruz & Wishart, 2017) to dermatology (Rajkomar, Dean & Kohane, 2019). Since COVID-19 pandemic has been affecting the global population for more than two years now and it is the cause of an immense health crisis in most world countries new diagnostic tools–machine learning being one of them–and therapeutic methods have been rapidly emerging (Alimadadi et al., 2020). Shortly after the COVID-19 outbreak various machine learning techniques were used, including taxonomic classification of COVID-19 genomes (Randhawa et al., 2020), determining the predictors of severe COVID-19 (Yan et al., 2020) and searching for new potential drug candidates against SARS-CoV-2 viral infection (Ge et al., 2021). Another example of a successful implementation of artificial intelligence in COVID-19 diagnosis is the evaluation of the CT scans detecting SARS-CoV-2 associated pneumonia and their differentiation from the community acquired pneumonia and other similar conditions with specificity and sensitivity higher than 90% (Li et al., 2020).

So far, several studies have been published using Random Forest Machine Algorithm for identifying the predictors for COVID-19 outcome from a wide variety of symptoms, socioeconomical factors (Wollenstein-Betech et al., 2020) and laboratory results with various results (Iwendi et al., 2020; Jie et al., 2020). To our current knowledge there are no studies specifically focused on the gastrointestinal symptoms and gut related laboratory findings to this date.

In order to assess the predictive power of the gastrointestinal parameters and other measured variables for predicting the need for hospitalization the Random Forest machine learning algorithm was trained on the data from our study. Random Forest has become the Machine Learning method of choice for several reasons: (a) it usually appears among the top performing classification algorithms; (b) it has small number of tuning parameters; (c) it does not overfit; (d) and last but not least, by its construction it provides a realistic estimate of the performance on a future data via the Out-Of-Bag data. Moreover, Random Forest, at least as implemented in the R library randomForestSRC, provides two different measures of importance of predictors. For these reasons, we have selected RF algorithm to assess predictive power of the studied variables, and to obtain their ranking.

Results were plotted as a ROC curve obtained from the Out-Of-Bag data. When considering the general COVID-19 symptoms, gastrointestinal symptoms, age, sex, lasting of the symptoms and comorbidities (diabetes mellitus, arterial hypertension and chronic liver diseases) the AUC is 0.76. The variable importance plot is shown in Fig. 5. When measuring the variable importance, the most important predictor is AST followed by age and diabetes mellitus, which are substantially less important. When using only liver enzymes (AST, ALT), gastrointestinal symptoms (diarrhea and bloating), age and presence of chronic liver disease and diabetes mellitus the AUC is 0.799 with AST as the strongest predictor for hospitalization. The variable importance plot is shown in Fig. 6. Previously published studies, which used mostly the methods of classical statistics, have identified the presence of gastrointestinal symptoms (Sun et al., 2020), predominantly diarrhea (Aumpan, Nunanan & Vilaichone, 2020; Ghoshal et al., 2020) and elevated liver enzymes (Aziz et al., 2020) as predictors of hospitalization associated with COVID-19. In our data, we have singled out aspartate transaminase (AST) as not only the statistically significantly elevated liver enzyme in patients requiring hospitalization, but using the artificial intelligence with the Random Forest algorithm the AST proved to be the most important predictor of hospitalization. Finally, we performed the principal component analysis for mixed type of data in order to obtain a two-dimensional representation of the data on patients who were discharged home and those who were admitted to hospital. As could be seen on the Plot 4 these two groups are partially overlapping, but with clear tendencies to shift apart, which is in accordance with the predictive performance of the studied variables in the Random Forest algorithm.

Figure 5 Variable importance plot for all measured factors.

Variable importance plot for all measured factors. The positive value of importance of a predictor represents a positive factor for the predictive accurancy of the Random Forest algorithm. The negative value of importance of a predictor indicates that omitting the predictor increases the predictive accuracy of the Random Forest algorithm.

Figure 6 Variable importance plot for selected factors.

Variable importance plot for selected factors that are fast and easy to measure in the emergency department setting (liver enzymes: AST and ALT, gastrointestinal symptoms /diarrhea and bloating/, age and presence of chronic liver disease and diabetes mellitus). The positive value of importance of a predictor represents a positive factor for the predictive accurancy of the Random Forest algorithm. The negative value of importance of a predictor indicates that omitting the predictor increases the predictive accuracy of the Random Forest algorithm.

Conclusions

This study has identified elevated AST for being the most important predictor for COVID-19 related hospitalizations using the machine learning Random Forest algorithm. We have also shown that SARS-CoV-2 positivity is connected with isolated AST elevation and the level is linked with the severity of the disease. Furthermore, the prevalence of diarrhea and nausea among SARS-CoV-2 positive patients is significantly higher compared to SARS-CoV-2 negative controls. Bloating is occurring significantly more frequently in COVID-19 patients who require hospitalization than those who could be discharged to outpatient care.

Supplemental Information

Supplemental Information 1 R code for the whole dataset, including random forrest algorithm and principal component analysis.

Libraries used in the code are cited in the methods and the sources in the bibliography section.

Click here for additional data file.

Supplemental Information 2 Raw data from emergency department.

1- in the means of “true” or “present”

0- in the means of “negative” or “not present”

Click here for additional data file.

Supplemental Information 3 Raw data from outpatient centre (for controls, means covid negative).

1- in the means of “true” or “prsenet”

0- in the means of “negative” or “not present”

Click here for additional data file.

Supplemental Information 4 Raw data from outpatient centre (for cases, means covid positive).

1- in the means of “true” or “presnet”

0- in the means of “negative” or “not present”

Click here for additional data file.

Supplemental Information 5 Questionnaire-English translation.

Click here for additional data file.

Supplemental Information 6 Questionnaire-original version.

Click here for additional data file.

Abbreviations

PCR polymerase chain reaction

ALT alanin transaminase

AST aspartate transaminase

GMT gamma glutamyltransferase

ICU intensive care unit

ED emergency department

ROC receiver operating characteristic

PCA principal component analysis

RF Random Forest

ATB antibiotics

Additional Information and Declarations

Competing Interests

Author Contributions

Human Ethics

Ethics

Data Availability

The authors declare that they have no competing interests.

Peter Lipták conceived and designed the experiments, performed the experiments, analyzed the data, prepared figures and/or tables, and approved the final draft.

Peter Banovcin conceived and designed the experiments, prepared figures and/or tables, and approved the final draft.

Róbert Rosoľanka performed the experiments, authored or reviewed drafts of the paper, and approved the final draft.

Michal Prokopič performed the experiments, authored or reviewed drafts of the paper, and approved the final draft.

Ivan Kocan performed the experiments, authored or reviewed drafts of the paper, and approved the final draft.

Ivana Žiačiková performed the experiments, analyzed the data, authored or reviewed drafts of the paper, and approved the final draft.

Peter Uhrik performed the experiments, authored or reviewed drafts of the paper, and approved the final draft.

Marian Grendar analyzed the data, prepared figures and/or tables, and approved the final draft.

Rudolf Hyrdel conceived and designed the experiments, authored or reviewed drafts of the paper, and approved the final draft.

The following information was supplied relating to ethical approvals (i.e., approving body and any reference numbers):

This study was approved by the Ethical committee of the University hospital in Martin, decision number: 14/2021.

The following information was supplied relating to ethical approvals (i.e., approving body and any reference numbers):

This study was approved by the Ethical committee of the University hospital in Martin, decision number: 14/2021.

The following information was supplied regarding data availability:

The raw data and the R code are available in the Supplemental Files.

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
