# Peer review of "A machine learning approach for identification of gastrointestinal predictors for the risk of COVID-19 related hospitalization"

_PeerJ, doi:10.7717/peerj.13124_

## Round 0.1 · original submission · Major Revisions

Your manuscript has been reviewed and requires several modifications prior to making a decision. The comments of the reviewers are included at the bottom of this letter. Reviewers indicated that the methods, results, and discussion sections should be improved. Reviewer 2 also recommended extensive English editing. I agree with the evaluation and I would, therefore, request for the manuscript to be revised accordingly. In addition to all these, please pay attention to the following points:

• Please give details for statistical methods in the methods section of the abstract.
• Provide p-values in the results section of the abstract.
• Line 152-157: It may be better to provide not only n but also % of the patients.
• In the literature, there are various machine learning algorithms that have been widely used recently and have higher performance values. It should be discussed why the random forest algorithm was chosen.
• Criteria values related to the performance of the random forest algorithm should be given, and the values obtained from the study should be compared with the relevant literature.

Reviewer 1 ·

Basic reporting

Rephrasing of the sentences needed in several places
e.g. line 96 - "2 distinct kinds of population had been considered for this study. First group consist of..." could be "Two distinct populations were considered for this study. The first group consisted of..."

Experimental design

Methods are not described with sufficient detail & information to replicate.
This reviewer found two main issues
1. The choice of machine learning before considering classical statistical models.
Although the authors correctly state that " Machine learning helps to quickly identify patterns and trends of the large volume of data, that are difficult for humans to recognize". However it seems that the number of variables and observations in the study are small enough for statistical models or the more interpretable machine learning approaches like lasso, ridge regression or other penalized approaches. Statistical models would also have helped in judging the direction, magnitude and significance of the effects.
It is understandable that various machine learning algorithms have been implemented in this field but we also need to justify the choice of the less interpretable models before proceeding to apply them.

2. It is not clear which specific variables were used for each of the analytical and visualisation approaches in the methods section "Data analysis". The reader cannot also clearly make a connection between each of the methods and variables in the results section. For instance, which variables were involved in the Chi-squared and Fisher test? Which null hypothesis and which of the factors were tested?
Which variables were tested using the multiple comparisons with the Benjamini Hochberg adjustment? Which variables were used in the contingency table, which continuous variables were tested
using the Kruskal Wallis test?
Rephrase the two-way ANOVA description (lines 139-141): the sentence is confusing and it is not clear which two factors and which interactions because multiple factors are mentioned here.

What were the predictors used for the machine learning models, 2D representation of the data by PCA?

Though this reviewer is conversant with R and could dig for this information in the supplementary files provided, readers not conversant with this analytical tool cannot relate to the analysis and results of this study in the absence of a clear connection between the variables and the analysis methods. This also applies for which methods were used to obtain which conclusions in the results section.

Validity of the findings

All underlying data have been provided; they are robust, statistically sound, & controlled

·

Basic reporting

The manuscript needs major polishing in terms of spelling, grammar, formatting and overall use of language. The presentation has some flaws in these aspects.

Experimental design

The experimental design is acceptable. The authors have used both statistical analysis and machine learning ( Random Forest ).

However, the authors may want to add other measure to test the accuracy of the model, compared to just ROC.

Also, the authors may want to compare the results using other machine learning algorithms such as decision tree, neural networks, extreme boost, logistic regression, and support vector machine.
At least the accuracy can be compared.

Validity of the findings

The findings seems fine.

An additional figure for Variable Importance would have been valuable.
Authors may also want to add a table containing all the variables with description.

Additional comments

In Discussion ( line 258 onwards ) seems more like Introduction. In Discussion, authors should compare their results with other related studies.

·

Basic reporting

The manuscript is aimed to explain the SARS-Cov-2 predictive factors considering gastrointestinal and liver problems as symptoms for the hospitalizations. The paper is written in a clear and correct language. However, there are fundamental problems as explained below.

Experimental design

no comment

Validity of the findings

1) It is not wise to factor out a dominant predictor. Clustering the factors, for example, considering two or more factors together in different combinations may provide better results than factoring out a single dominant predictor. Plus, the difference of the ROC obtained is, as outlined in lines 188-190, only 3%, which is very small to arrive at such bold conclusion.
2) The difference in patients admitted to hospital and discharged home for AST is 0.74 – 0.56 = 0.18, for ALT, it is 0.61-0.48 = 0.13. This implies the mean values are almost equal showing that AST is not conclusive enough to factor it as an important predictor than ALT.
2) Most of these symptoms are also dependent on other environmental and epidemiological factors unique to a specific type of patient. So, there is no preferred predictor for this specific type of disease.
3) Some of the points that are inconclusive based on available data but mentioned in the paper. For example, on Line 161, and Table 1 the total number of female and male patients considered is 347 as opposed to 352 as explained in the “detailed comments” section below.

Additional comments

Considering, the above concerns and the following detailed comments, the paper needs to carefully address these issues before it is considered for publication.

Detailed comments:

Line 96: “2 distinct …” may be re-written as “Two distinct…”. Starting sentences with numbers may create confusion since the authors also used numbers for citations.

Table 1: In the “Gender” row and “Outpatient test center: SARS-CoV-2 negative” category, the number of female, 206 and male, 141 add up to 347 but the number, n of the total negative outpatient number is 352. What is the reason for this discrepancy?
Line 161: “There were no significant differences based on sex.” The data isn’t shown for the gender category. So, how do we know this?

Lines 166-168: “Comparing SARS-Cov-2 negative and SARS-CoV-2 positive participants the presence of these symptoms has been more than three times higher in the positive group than in the negative one.” This sentence doesn’t seem to be supported by data. According to Table 1: The number of positive group, who are presented with the symptoms are 122 and the negative 101. They are almost equal and the difference is NOT a multiple of three.

Lines 188-190: The AUC values for the two curves is 0.799 and 0.76, the difference of which is 0.033 = 3%. So, can these be conclusive enough to quantify the dominance of the liver enzymes? Also as explained here,
Lines 59 – 61: “Furthermore, using machine learning random forest algorithm, we have identified elevated AST as the most important predictor for COVID-19 related hospitalizations.”

---

## Round 0.2 · Minor Revisions

The manuscript has been assessed by three reviewers, and one of them agrees with the fact that there are still a few points that need to be addressed. We would be glad to consider a substantial revision of your work, where the reviewer’s comments will be carefully addressed one by one.

Reviewer 1 ·

Basic reporting

No comment

Experimental design

No comment

Validity of the findings

No comment

Additional comments

Authors have adequately addressed issues raised during the first review

·

Basic reporting

The ms is now much better compared to the first draft.

Experimental design

Acceptable

Validity of the findings

Acceptable

Additional comments

The authors addressed the comments that were raised.

·

Basic reporting

In this revision, the authors tried to incorporate some of the comments and provided rebuttal for questions that required explanations. This revised version is better than the original article but there are still some problems. Such as, (a) language, abbreviation usage, and punctuation still require editing, (b) the graphs occupied unnecessarily large spaces and on two pages. These figures can be minimized to occupy a single page,, (c ) the Figure captions are not explanatory enough as explained in the detailed comments below, and (d) the comments mentioned in the detailed comments below need attention.

Experimental design

no comment

Validity of the findings

no comment

Additional comments

Lines 176-177: “...O2 saturation, CRP, GMT, AST, Bilirubin…). The abbreviations are mentioned for the first time. These have to be interpreted.
Line 180: ATB is mentioned for the first time without interpretation.
Line 200: “ 30 patients were …” Starting sentences with numbers are still present in the manuscript even though the authors corrected some of the paragraphs in the introduction.
Line 227: “... wasalso …” is written as one word. Consider correcting it.
Lines 227-231: The figures for ALS and AST are separate and it is difficult to compare. Perhaps combining the two graphs may provide clear information. Plus there is no figure for Bilirubin but in the statement, “ There is no statistically significant difference in the levels of ALT(Figure 2) and Bilirubin.”
Lines 247-252: Figure 5 has two dimensions dim1 (~14%) and dim2(~10%). These are to indicate the first pca and second pca respectively. Neither the explanation nor the Figure captions has sufficient explanation about the PCA values.

---

## Round 0.3 · accepted · Accept

Dear Dr. Lipták and colleagues:

Thanks for revising your manuscript based on the concerns. I now believe that your manuscript is suitable for publication. Congratulations! I look forward to seeing this work in print. Thanks again for choosing PeerJ to publish.

Best,
Aslı

·

Basic reporting

No comment

Experimental design

No comment

Validity of the findings

No comment

Additional comments

Almost all the comments were addressed.